# MerTK and the Role of Phagoptosis in Neonatal Hypoxia-Ischemia

**DOI:** 10.3390/cells14231862

**Published:** 2025-11-26

**Authors:** Andrea Jonsdotter, Henrik Hagberg, Anna-Lena Leverin, Joakim Ek, Kerstin Ebefors, Eridan Rocha-Ferreira, Ylva Carlsson

**Affiliations:** 1Department of Obstetrics and Gynecology, Sahlgrenska University Hospital, 416 85 Gothenburg, Västra Götland, Sweden; henrik.hagberg@obgyn.gu.se (H.H.); ylva.carlsson.2@obgyn.gu.se (Y.C.); 2Centre of Perinatal Medicine & Health, Institute of Clinical Sciences, Sahlgrenska Academy, University of Gothenburg, 405 30 Gothenburg, Västra Götland, Sweden; anna-lena.leverin@neuro.gu.se (A.-L.L.); joakim.ek@neuro.gu.se (J.E.); eridan.rocha.ferreira@gu.se (E.R.-F.); 3Centre of Perinatal Medicine & Health, Institute of Neuroscience and Physiology, Sahlgrenska Academy, University of Gothenburg, 405 30 Gothenburg, Västra Götland, Sweden; 4Department of Physiology, Institute of Neuroscience and Physiology, Sahlgrenska Academy, University of Gothenburg, 405 30 Gothenburg, Västra Götland, Sweden; kerstin.ebefors@neuro.gu.se

**Keywords:** brain injury, hypoxic-ischemia, MerTK, mouse, neonatal, phagocytosis

## Abstract

**Highlights:**

**What are the main findings?**

**What are the implications of the main findings?**

**Abstract:**

Brain damage caused by hypoxia-ischemia is a serious complication for a newborn with possible life-long sequelae. To develop targeted neuroprotective strategies, it is essential to understand the mechanisms of injury, particularly the role of microglial phagocytosis, which may contribute to neuronal loss after hypoxia-ischemia. The aim was to evaluate neuronal cell death by phagocytosis in neonatal hypoxia-ischemia by investigating key signaling molecules and the effect of gene deletion of the phagocytic receptor Myeloid-epithelial-reproductive tyrosine kinase (MerTK) in a neonatal mouse model. MerTK, growth arrest–specific 6, and genes related to phagoptosis were regulated in the brain 6–72 h after hypoxic ischemia. Brain injury was reduced in MerTK knock-out vs. wild-type mice by 48% in gray matter (*p* = 0.002) and by 32% in white matter (*p* = 0.04). There was a near 40% reduction in NeuN immunoreactivity in microglia in MerTK knock-out mice vs. wild-type (*p* = 0.03) indicating attenuation of neuronal phagocytosis by microglia. In summary, the reduction in microglial neuronal engulfment and brain injury in MerTK-deficient mice strongly indicates that phagoptosis contributes to neuronal loss after neonatal hypoxia-ischemia. This insight suggests that targeting MerTK-mediated phagocytosis may represent a potential therapeutic approach in neonatal hypoxia-ischemic brain injury.

## 1. Introduction

Perinatal brain damage caused by hypoxia-ischemia (HI) remains a major global health problem and a leading cause of neonatal encephalopathy and death in term infants [1]. Although therapeutic hypothermia improves survival, significant neurological morbidity persists, and many infants worldwide, including those born preterm, are not eligible for this treatment [2,3]. Therefore, there is still an urgent need to understand the underlying pathophysiology and find new strategies for brain protection [4]. The cellular mechanisms behind cell death after brain injury are not yet fully understood. We and others have previously shown that microglia are markedly activated after neonatal HI [5,6], contributing to the proinflammatory response including inflammasome activation and cytokine production [7]. Microglia are also considered to be the main phagocytosing cell in the central nervous system (CNS) [8], but it is uncertain whether phagocytosis has adverse or beneficial effects. Phagocytosis is a well-established process whereby debris and dead cells are engulfed and degraded [9]. Thereby the inflammatory process is attenuated and the ability to regenerate and remodel synaptic, dendritic and axonal networks is improved. However, recent data in adult ischemia indicates that phagocytosis can contribute to the death of viable neurons that are stressed but not injured irreversibly [10], a process named cell death by phagocytosis or phagoptosis [10,11]. Previous work has demonstrated that inhibition of microglial phagocytosis after brain ischemia or lipopolysaccharide (LPS) induced dopaminergic cell loss in adult animals can rescue neurons from dying and attenuate brain injury [12,13].

There are several chemoattractants, opsonins, and phagocytic receptors suggested to be involved in phagoptosis (Figure 1) [8,11]. The Myeloid-epithelial-reproductive tyrosine kinase (MerTK) receptor is a phagocytic receptor located on microglia, and is activated through opsonins growth arrest–specific 6 (Gas-6), complement factors or osteopontin, which all can bind to the “eat-me signal” phosphatidylserine (PS) or sialoglycans exposed on the outer layer of the cell membrane on damaged neurons, and then phagocytosis is mediated by microglia [11,14]. Brown et al. showed that in adult mice MerTK was upregulated after focal ischemia, with levels peaking 72 h after induced focal ischemia. By deleting the MerTK gene (with wild type (WT) animals serving as controls), they were able to demonstrate a notable decrease in brain atrophy following brain ischemia, ultimately resulting in improved motor outcome [10].

To our knowledge, the role of phagoptosis and MerTK has not yet been investigated in neonatal HI but since cell death by phagocytosis is heavily involved in development of the immature brain [15], the molecular machinery necessary for phagoptosis is in place at an early age and is anticipated to be activated in response to HI in the neonatal brain. This study aimed to evaluate to what extent the molecules critical for phagoptosis are expressed and to assess the potentially neuroprotective effect of MerTK gene deletion in neonatal mice after HI.

## 2. Methods

### 2.1. Animals

The mice were of the C57BL/6 strain, procured from Charles River Laboratories. For the KO animals, gene-modified B6;129-MerTK tm1Grl/J mice were employed, and they were cross-bred with the WT animals of the 101045 B&129SF2/J Jackson strain (Jackson Laboratory, Bar Harbor, ME, USA). Litters with mixed genotype (MerTK^+/+^, MerTK^+/−^, MerTK^−/−^) were exposed to HI and genotyping of each animal was performed at the time of euthanasia. DNA from tails or ears was prepared using Sigma reagents (Sigma-Aldrich, St. Louis, MI, USA) and their protocol; Extraction Solution (E7526), Tissue Preparation Solution (T3073) and Neutralization Solution B (N3910). PCR was preformed using GoTaq G2 Hot Start Green Master Mix (Promega, Madison, WI, USA) and primers (mutant Forward: ATC AGC AGC CTC TGT TCC AC, wild type Forward: TGC CAT TAT ACC TGG CTT TCA and Common Reverse: CAT CTG GGT TCC AAA GGC TA). PCR was run with 40 cycles, denaturation at 95 °C 20 s, annealing at 60 °C 15 s, and elongation at 72 °C 20 s; samples were separated on an 2% Agarose/TBE gel containing SYBR™ Safe DNA Gel Stain (Invitrogen, Waltham, MA, USA) and checked in a CCD camera. Mutants (KO) received a 250 bp band, Wild type (WT) a 235 bp band and heterozygotes received both bands. All experiments conformed to the Swedish Board of Agriculture and were approved by the Gothenburg Animal Ethics Committee (62-2016 and 002195). All animals were housed under standard laboratory conditions with free access to food and water. Pups of both sexes were used for all experiments. Gender was evenly distributed. The ARRIVE guidelines were followed throughout the study.

### 2.2. Hypoxia-Ischemia

The Vannucci model [16] of HI modified for mice was applied as previously described [5]. HI was induced on postnatal day (PND) 10. The left common carotid artery was ligated through a 3 mm long skin incision under anesthesia with isoflurane, 5% for induction and 3% for maintenance; the mice were then returned to their dam to recover for one hour. Body temperature was not measured during this brief (5 min) period of surgery. Thereafter the animals were placed in an incubator with a constant temperature of 36 °C and 10% O_2_ for 50 min. After HI, the animals were returned to their dams until euthanasia. The mortality rate was low (<1%) both during surgery and throughout the period until the animals were euthanized for analysis. In C57BL/6 mice, the protein expression of MerTK and Gas-6 were investigated at 6 h, 12 h, 24 h, 72 h, and 7 days after HI. We also included naive controls (6 animals/group), killed at PND10, 11, 13, and 16.

In gene modified animals both WT and knockout (KO) mice were used from the same litter. WT (*n* = 20) and MerTK KO mice (*n* = 20) were exposed to HI and killed 7 days later for assessment of brain injury in brain sections stained for the neuronal marker microtubule-associated protein-2 (MAP-2) and white matter marker Myelin Basic Protein (MBP). Protein analysis and activity measurements (Caspase-3 activity, Gas-6 ELISA, Cytokine/chemokine Bio-plex assay, Western blot: P (phosphorylated)-STAT3 were performed at 6 h after HI (13 WT and 10 KO) as well as 72 h after HI (10 WT and 10 KO). Unless otherwise specified, all histological and immunohistochemical evaluations were consistently performed at coronal levels corresponding to the striatum and the thalamus/hippocampus, which are the regions that were most reproducibly affected in this neonatal HI model.

### 2.3. RNA Sequencing

Total RNA was prepared from brain tissue (*n* = 6, MerTK KO and WT naive PND10, 11 and 13 days and MerTK KO and WT 6 h, 24 h and 72 h after HI) by using miRNeasy Mini kit (Qiagen, Solna, Sweden). Total RNA concentration and purity was determined with a Nanodrop. RNA quality check and mRNA Next Generation Sequencing were performed by Qiagen (Hilden, Germany).

Library preparation and sequencing was performed using the QIAseq Stranded mRNA Kit (QIAGEN). From 500 ng RNA, the mRNA was enriched. The RNA was heat fragmented. After first and second strand synthesis, the cDNA was end-repaired and 3′ adenylated. Sequencing adapters were ligated to the overhangs. Adapted molecules were enriched by 13 cycles of PCR and purified by a bead-based cleanup. Library preparation was quality controlled using capillary electrophoresis (Agilent DNA 1000 Chip, Santa Clara, CA, USA). High quality libraries were pooled based in equimolar concentrations. The library pool(s) were quantified using qPCR and optimal concentration of the library pool were used to generate the clusters on the surface of a flowcell before sequencing on a NextSeq (Illumina Inc., San Diego, CA, USA).

### 2.4. Immunohistochemistry

#### Tissue Preparation

At PND13 (72 h after HI) and PND17 (7 days after HI), mice were deeply anesthetized by intraperitoneal injection of (i.p.) thiopental (Pentocur, Thiopental, 50 mg/mL, Abcur AB, Helsingborg, Sweden) and then decapitated. The brains were dissected out and fixed in paraformaldehyde (Histofix; Histolab Gothenburg, Sweden), dehydrated and embedded in paraffin and cut with a microtome (Meditome A550, MEDITE Medical, Burgdorf, Germany) into 7 μm sections throughout the whole brains. Brain sections were deparaffined in xylene followed by alcohol rehydration and boiling in citric acid buffer (0.01 M, pH6) for antigen recovery.

The brain sections (7 days after HI) were then treated and blocked for endogenous peroxidase with H_2_O_2_ in phosphate-buffered saline (PBS). Brain sections were incubated with primary antibody overnight (MAP-2 1:1000, M4403, Sigma, Saint Louis, MO, USA and MBP 1:1000, #836504, BioLegend, San Diego, CA, USA) followed by incubating in horse-anti mouse biotinylated secondary antibody (BA-2001, Vector, Newark, CA, USA) for 1 h and in ABC (PK-6100, Vector) solution for 1 h. Finally, the sections were stained in 3,3-diaminobenzidine-solution, dehydrated, and mounted with cover slips.

### 2.5. Brain Injury Evaluation

Brain injury in the gray matter MAP-2 and white matter MBP (corpus callosum/external capsule) were calculated as the mean area tissue loss at eight levels throughout the brain using every 50th cut section. The images were taken by microscope Olympus BX60F5 and the total area of MAP-2 and MBP positive staining were measured in ImageJ software (version 1.53t, NIH) and the amount of area tissue loss was calculated by subtracting the positive area in the ipsilateral hemisphere from the contralateral one expressed as % of the contralateral hemisphere. The analysis of brain injury after different tissue staining’s were performed without knowledge of genotype by two separate investigators.

### 2.6. NeuN/Isolectin

Brain sections (72 h after HI) were blocked with Dako block (X0909) 30 min at room temperature (rt). The brain sections were then incubated with NeuN antibody (MAB377, Merck Millipore, Burlington, MA, USA) 1:250 in PBS containing 4% donkey serum, 0.2% triton-X100 for 2 h at rt followed by incubation with secondary antibody donkey-anti-mouse-594 (A21203, Alexa Fluor, ThermoFisher, Waltham, MA, USA) 1:250 in PBS. The sections were then incubated with FITC-labeled isolectin (L0401, Sigma) 10 µg/mL (1:200) in PBS, 2 h at rt. Finally, the sections were rinsed in H_2_O and mounted with Prolong Gold (ThermoFisher). The colocalization of isolectin (microglia) and NeuN (Neurons) was used to assess engulfment of neurons by microglia. The number of isolectin^+^-cells with or without co-staining with NeuN was counted, with DAPI nuclear staining used to distinguish cells from the surroundings, in two areas of the striatum (450 µm × 350 µm) at approximately the level of Bregma. The analysis was performed by two individuals blinded to grouping of animals and results averaged.

### 2.7. Confocal Microscopy

Immunofluorescence staining was visualized in striatum both in MerTK KO and WT animals, using an Axio Imager.Z2 LSM800 confocal microscope with software zen blue version 3.7 (Carl Zeiss, Oberkochen, Germany), at 20× or 40× magnification with 3 times digital zoom. Z-stacks were acquired and used for 3D visualization and for 2D pictures through post processing to orthogonal projection images. For 2D, a minimum of 10 images and for 3D, 30 images at different levels in the sections were acquired.

### 2.8. Protein Samples Preparation

Brain tissue was homogenized and sonicated in PBS-buffer containing 1% PhosSTOP phosphatase inhibitor cocktail (tablets Roche Diagnostics, Rotkreuz Switzerland), 1% Protease Inhibitor Cocktail (P8340, Sigma), 5 mM EDTA, 1% Triton X-100. The samples were centrifuged for 10 min at 10,000× *g* in 4 °C. Protein concentration was quantified in the supernatant by BCA assay (ThermoFisher Scientific).

### 2.9. ELISA

ELISA of Mer-TK (ab210572, Abcam, Cambridge, UK) and Gas-6 (ab155447, Abcam) was analyzed. Protein samples from C57BL/6 mouse brains, *n* = 6/group (naive controls PND9, PND10, PND12, PND16 and contralateral and ipsilateral hemispheres at 6 h, 24 h, 72 h, and 7 days after HI), were diluted to 1 mg/mL in appropriate buffers from the two ELISA kits and were analyzed according to Abcam protocols.

### 2.10. Western Blot

Protein samples were prepared for TGX stain-free Criterion gels (Bio-Rad Laboratories GmbH, Munich, Germany) according to Bio-Rad. Denatured protein, 30 μg, were loaded and separated on 26-well 4–20% TGX stain-free gels and transferred to a Nitrocellulosa membrane (Bio-Rad). Photos were taken on the membranes after ultraviolet activation of the Criterion stain-free gel in the ChemiDoc MP Imaging System (Bio-Rad). The membranes were blocked by 5% *w*/*v* non-fat dry milk in TBS-T for 1 h at rt. The membranes were then incubated with different antibodies (Protein-S 1:250 MAB4976 R&D System, Minneapolis, MN, USA and P-STAT3 1:2000 #9145, P-Erk 1:2000 #4370, P-Akt 1:2000 #4060 Cell Signaling Technology, Danvers, MA, USA) in TBS-T containing 3% *w*/*v* Bovine Serum Albumin, followed by a HRP conjugated secondary antibody (Vector) and incubation with Super Signal West Dura Extended Duration Substrate (ThermoFisher Scientific). The chemiluminescence signal was captured using a ChemiDoc MP Imaging System (Bio-Rad). Band densitometry and quantification were performed using Image Laboratory (Version 5.0, Bio-Rad, Sweden), and the protein band densities were normalized to the total protein measurements in each sample.

### 2.11. Bio-Plex

Protein samples, 1 mg/mL, were run with a 7-Bioplex assay for interleukin-1beta (IL-1β), tumor necrosis factor-alpha (TNF-α), interleukin-6 (IL-6), interleukin-10 (IL-10), monocyte chemoattractant protein-1 (MCP-1), keratinocyte chemoattractant (KC) and gamma-induced protein 10 (IP-10) according to Bio-Rad instructions. Cytokines and chemokine concentrations were calculated from standard curves from the same run.

### 2.12. Caspase-3 Activity Measurement

Caspase-3 activity was measured using a standard fluorometric DEVD-AMC assay according to the manufacturer’s instructions. Fluorescence was recorded at 380/460 nm at 37 °C, and activity was expressed as picomoles AMC released per minute per milligram of protein.

### 2.13. Statistical Analysis

Statistical analyses were performed in GraphPad Prism 9. Parametric tests (unpaired *t*-test) were used for normally distributed data, and non-parametric tests (Mann–Whitney or Kruskal–Wallis with Dunn’s post hoc test) were used otherwise. A *p*-value ≤ 0.05 was considered statistically significant.

Differential gene expression analysis was performed in DESeq2 with Benjamini–Hochberg correction (FDR < 0.05). Gene Ontology enrichment was conducted using ShinyGO, and PCA plots and heatmaps were generated in Qlucore Omics Explorer (version 3.9).

## 3. Ethics

All animal experiments were in line with the guidelines established by the Swedish board of Agriculture and approved by Gothenburg Animal Ethics Committee (nr 62-2016 and 002195), date of approval: 29 June 2016 and 3 April 2019.

## 4. Results

### 4.1. Gene Expression Related to Microglial Phagoptosis

RNA sequencing performed on left hemisphere of WT mice showed that many genes coding for signaling proteins that were previously shown to be important in phagoptosis were regulated following HI (Figure 2) [11].

Many so-called chemoattractants were expressed such as the purinergic receptor P2y12 [17], fractalkine, and its receptor [18], cxcl8 mouse analog gro1 [19], spingosine-1-phosphate receptor (s1Pr2) [20], complement 3a and 5a and their receptors [21]. Next, we explored genes related to “eat me signaling” (Figure 1) [8,11]. We generated PCA-plots showing that these genes are regulated in a distinct time-dependent manner after HI, whereas only small changes were seen in control animals over time (Figure 3). Furthermore, genes for “eat me” signaling proteins shown to be important in cell death by phagocytosis were regulated including opsonins such as Gas-6, osteopontin (spp1) [22], C1q [23], complement 3, and the phagocytic receptors TREM2 [24], CR3 [8], MerTK and p2y6 (P2ry6) (Figure 2) [25]. The gene expression of the scramblase TMEM16F (ano6) responsible for phosphatidylserine exposure on neurons [26] was also increased after HI.

### 4.2. Protein Expression of MerTK and Gas-6

The phagocytic receptor protein MerTK expression in C57Bl/6 mice exposed to HI was initially lower than in control animals at 6 h post HI (*p* ≤ 0.01), but at 24 h post-HI the MerTK expression was significantly upregulated (*p* ≤ 0.01) when compared to naïve control animals. A similar trend was found at 72 h and 7 days, but the difference did not reach statistical significance.

The phagocytic opsonin Gas-6 protein expression in C57Bl/6 mice tended to be upregulated initially at 6 h, and the expression was significantly higher in the HI group compared with controls at 24 h (*p* < 0.002), 72 h (*p* ≤ 0.004), and 7 days (*p* ≤ 0.002) (Figure 4).

### 4.3. Brain Injury in MerTK KO and WT Animals After HI

Deletion of the gene for the phagocytic receptor MerTK was associated with a marked reduction in the extent of brain damage 7 days after HI. The total mean area loss in gray matter (MAP-2) was reduced from 31.1 ± 3.1% (mean ± SEM) in MerTK WT to 16.2 ± 2.3% in MerTK KO mice (*p* = 0.002; Figure 5A,C). Brain injury in MerTK^+/−^ heterozygotes (32.4 ± 4.5 %; *n* = 20) did not differ from that in WT mice. The total white matter loss (MBP) was decreased from 27.6 ± 3.7% in WT to 18.9 ± 2.3% in MerTK KO mice (*p* = 0.04; Figure 5B,D). There was no difference in HI brain injury between male and female mice.

### 4.4. Analysis of Microglial Phagocytosis of Neurons After HI

Co-staining of the neuronal marker NeuN and the microglial marker isolectin using immunofluorescence was performed in striatum 72 h after HI, a region where MerTK gene deficiency conferred neuroprotection (Figure 6). A significant (*p* = 0.03) reduction in the mean % microglia with NeuN staining in MerTK^−/−^ (11.0 ± 3.0) vs. MerTK^+/+^ (18.0 ± 2.4) mice was found. The results are displayed in (Figure 6) demonstrating NeuN staining in isolectin-positive microglia using confocal microscopy.

### 4.5. Gene and Protein Expression After Brain HI in MerTK KO and WT Mice

To better understand the consequences of deletion of the phagocytic receptor MerTK in neonatal HI, the global gene expression in MerTK KO and WT mice was compared in naïve animals and at 6 h, 24 h and 72 h after HI (Figure 2). We found marked differences in genes involved in inflammation (e.g., *Adnp*, *Eif2s3y*, *Xlr4b*, *Cpne1 Gm4737*, *Gbp4*, *Gatm, Aplnr*, *Pla2g4b*, *Pla2g4e*, *B2m*, *Ahcy*, *Rig-1*(*Robo3*), *Kcna1*, *Ptgds*) in KO vs. WT animals, but also genes related the Wnt/β-catenin pathway (*Kcnip3*, *Gprin3, Serpina3n*), ERK-pathway (*Eif3j2*), phagocytosis (*Npcd*, *Syt2*, *Ttr*, *Bloc1s6*, *Rasgrp1*, *Glra1*) and cell death (e.g., *Adnp*, *Gucy1a2*, *Casc4*, *Tmem87a*, *Prex2*, *jmjd7*) were differentially regulated as shown in a heat map (Figure 7).

The predominate differences in inflammatory pathways were also confirmed in a gene ontology (GO)-term analysis indicating chronic inflammation as one of the top biofunctions differently regulated in MerTK KO vs. WT animals (Figure 8).

To confirm the gene differences in the inflammatory pattern in MerTK KO compared to WT selected analysis of specific proteins was performed. The cytokines showed IL-6 and MCP-1 significantly reduced in KO mice 6 h after HI (*p* = 0.004 and *p* < 0.0001, respectively) and a marked reduction in P-STAT3 in KO vs. WT animals at 6 h after HI, *p* < 0.0001, (Figure 9). Also, caspase-3 activation 6 h after HI was reduced (*p* = 0.05) in KO mice matching the reduction in brain injury.

## 5. Discussion

This study shows that many genes believed to be involved in phagoptosis in the adult brain are also regulated after neonatal HI and that the phagocytic receptor protein MerTK and its opsonin binding partner Gas-6 are both upregulated after the insult. Furthermore, MerTK gene deletion reduced phagocytosis of neurons and markedly attenuated brain injury. Collectively, these results indicate that neuronal cell death by microglial phagocytosis occur also in the neonatal brain as previously reported in adult ischemia [12]. Gene and protein analysis revealed that the immuno-inflammatory response was different in MerTK KO animals which may contribute to the protective effect of MerTK gene deficiency. In addition to resident microglia, infiltration of peripheral macrophages may occur after HI. Although their specific contribution and receptor profile in this neonatal model remain unknown, such infiltrating phagocytes could potentially influence the overall phagocytic response.

Microglial phagocytosis after ischemia has been proposed to be part of the defense removing cellular debris and thereby inhibiting inflammation and promoting regeneration [27]. Recent reports suggest however that neuronal death can also occur as a result of microglial phagocytosis of stressed but not irreversibly injured neurons. This concept is supported by the fact that blocking or deleting specific phagocytic receptors (MerTK, P2Y6) or opsonins milk fat globulin-E8 (MFG-E8) can prevent neuronal loss in animal models of ischemic brain injury [8,11]. Presumably, the severity and type of the insult as well as which specific phagocytic mechanisms are manipulated will determine whether live neurons or merely irreversibly injured cells are engulfed.

Microglial phagocytosis requires inflammatory activation of microglia, conditions that stress neurons leading to the release of chemoattractants, expression of opsonins, and activation of phagocytic receptors [8]. Indeed, in neonatal HI there is a shift in microglia from a resident to an amoebic phenotype shown to produce proinflammatory cytokines and chemokines [28,29]. Neurons are metabolically challenged [30] and are exposed to toxic levels of glutamate [31] and reactive oxygen species [32] during the recovery phase after HI. We found presently that many genes of chemoattractant proteins (P2y12, fractalkine, gro-1, s1Pr2, c3a, c5a) are differentially regulated (Figure 2). The microglial receptor P2y12 appears to respond to nucleotides released from injured neurons and P2y12 deficiency confer neuronal protection [17]. Microglial phagocytosis of myelin can be mediated via the C3-C3R pathway and deletion of *C3ar1* significantly reduced white matter injury after hypoperfusion [21]. Also, genes for opsonins (besides Gas-6) believed to be involved in phagoptosis were regulated in the neonatal brain after the insult (Figure 2 and Figure 3). For example, osteopontin is produced by microglia after ischemia and the protein adhere to neuronal but not myelin components and facilitate phagocytosis [22] and C1q is an important opsonin in neuronal phagocytosis by microglia [23]. We also found that genes for several phagocyte receptors (besides MerTK) were regulated. TREM2 on both microglia and macrophages is important for phagocytosis [24]. UDP/ADP release from stressed neurons has been shown to provoke phagocytosis of viable neurons via the purinergic P2y6 receptor [10]. Interestingly, in (Figure 7) there is regulation of different genes in the KO control animals at P10 which hypothetically could provide a preconditioning effect and might confer neuroprotection in response to the subsequent HI insult. However, none of the genes regulated have previously be shown to be involved in preconditioning [33,34].

In the present study microglial phagocytosis of neuronal material was reduced and concomitantly brain injury was reduced in both gray and white matter. The proposed mechanism is that MerTK interact with the opsonin Gas-6 (both upregulated after HI, (Figure 4) bridging to phosphatidylserine exposed on the neuronal surface [8]. However, MerTK can also induce engulfment through a cross reaction with the vitronectin receptor (VNR) [12] or through interaction with other opsonins and binding partners (Figure 1). Whether the NeuN positive immunoreactivity was part of live neurons or merely cellular remnants was not investigated. It is difficult to determine if the protection offered by MerTK gene deficiency was entirely due to reduction in cell death by phagocytosis or by other mechanisms. For example, the pattern of the immuno-inflammatory response was different after HI in MerTK KO compared to WT mice, but it is difficult to tell if these differences contribute to the extent of injury or are merely a consequence of the marked difference in the extent of brain injury.

The different inflammatory pattern in MerTK KO mice could be due to the deletion of the phagocytic receptor and changes to the microglial phenotype [35] in such a way that repair and regeneration is promoted instead of phagocytosis which speculatively could contribute to brain protection. STAT3 has been shown to be involved in microglial phagocytosis [36,37] and the decrease in pSTAT3 could be secondary to suppression of the phagocytotic process in MerTK KO mice. Furthermore, several genes were found to be related to the Wnt/β-catenin pathway which was differently regulated in MerTK KO vs. WT animals. This is interesting since this pathway was previously found to be protective in a model of perinatal brain injury [38].

In conclusion, this study reports that many genes related to cell death by phagocytosis were regulated after neonatal HI. The phagocytic receptor protein MerTK and the ligand Gas-6 were upregulated after the insult and MerTK gene deficiency conferred significant neuroprotection and attenuated the microglial engulfment of neuronal components. The loss of the phagocytic MerTK receptor also modulated the inflammatory response and the expression of genes related to the Wnt/β-catenin pathway which speculatively may contribute to the neuroprotection.

## Figures and Tables

**Figure 1 cells-14-01862-f001:**
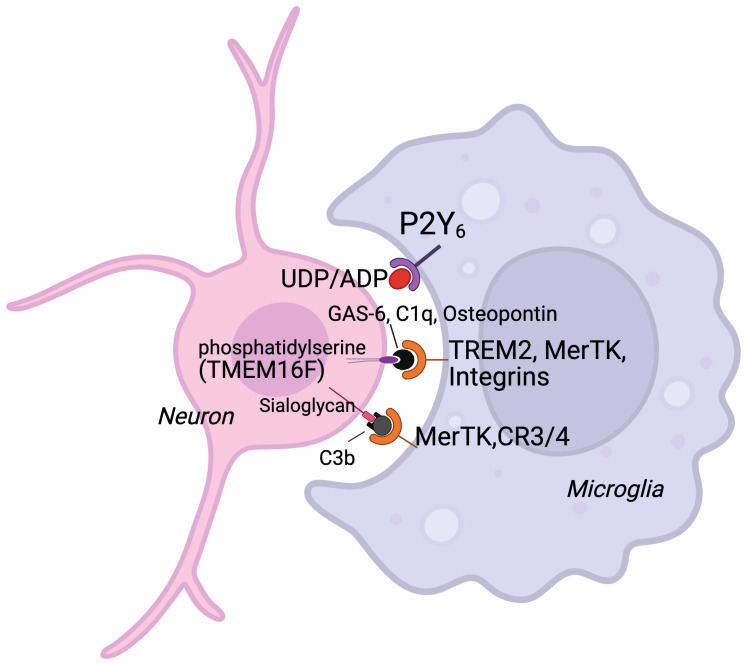
Selected signaling pathways in phagoptosis. Neurons that are stressed release ADP (Adenosine diphosphate) and UDP (Uridine diphosphate), and expose ‘eat-me’ signals such as phosphatidylserine and sialoglycan. Activated microglia attach and engulf neurons by expression of phagocyte receptors (MerTK (myeloid-epithelial-reproductive tyrosine kinase), P2Y6 (Purinergic receptor 6), TREM2 (Triggering receptor expressed on myeloid cells 2), integrins and CR3/4) which is facilitated by expression of opsonin molecules such as Gas-6, C1q, osteopontin and C3b. The scramblase TMEM16F (Transmembrane protein 16F) flips phosphatidylserine which is thereby exposed on the neuronal surface and can be recognized by the phagocytizing microglia.

**Figure 2 cells-14-01862-f002:**
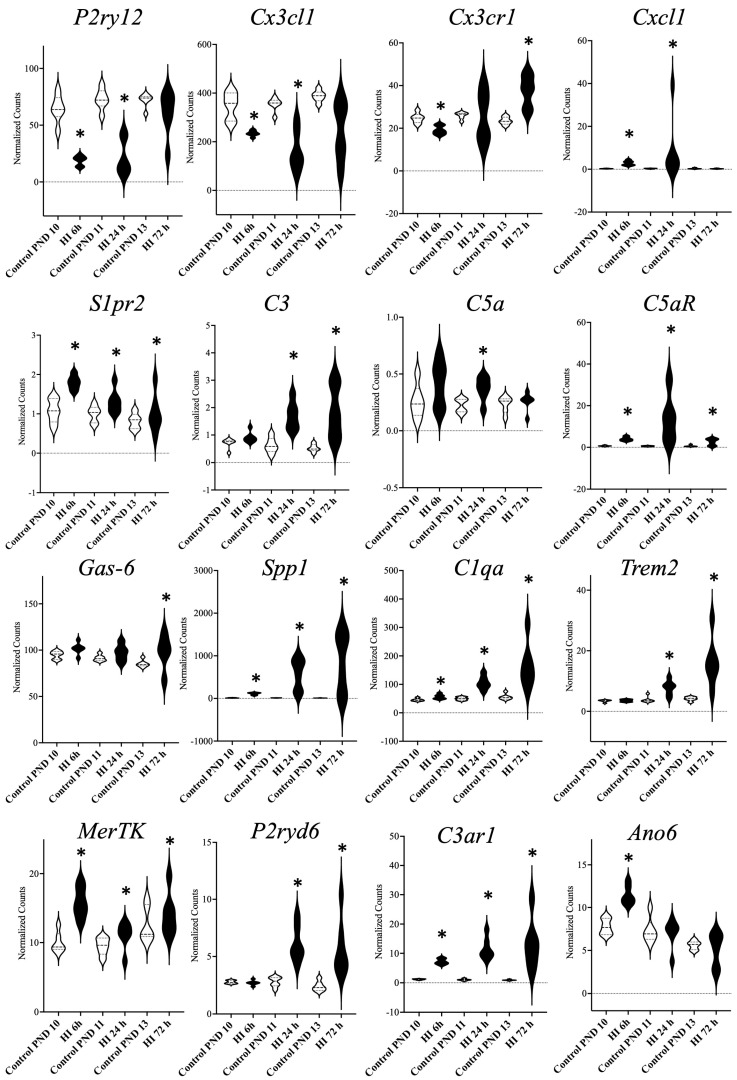
Phagoptosis related genes expressed after HI. Genes considered to be involved in phagoptosis that are differentially regulated at different timepoints (6 h (hour), 24 h and 72 h) after HI (hypoxic-ischemia) in WT (wild type) animals (HI), * FDR < 0.05.

**Figure 3 cells-14-01862-f003:**
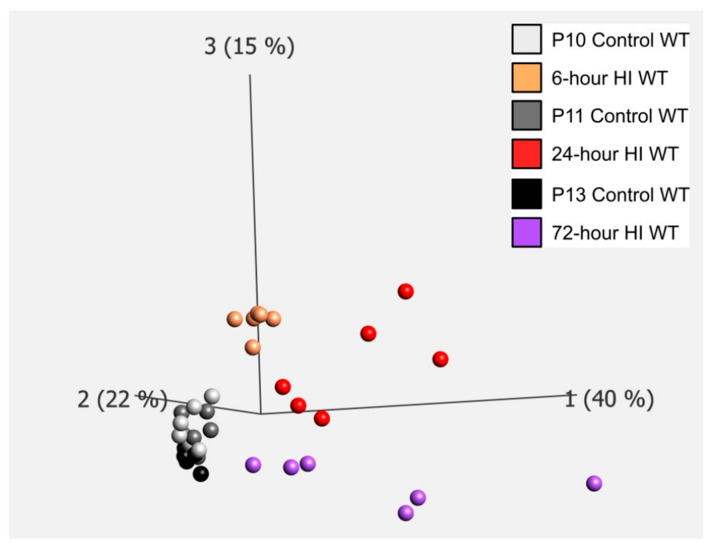
PCA plotting of ‘eat me signals’ in WT animals with and without HI. PCA—Principle Component Analysis demonstrating that expression of genes relating to “eat-me signaling” change at the different time points after HI (hypoxic-ischemia) as compared to controls. All control animals (white/gray/black) cluster together whereas HI animals (golden/reds/purple) are separated from control animals and cluster at their specific time point.

**Figure 4 cells-14-01862-f004:**
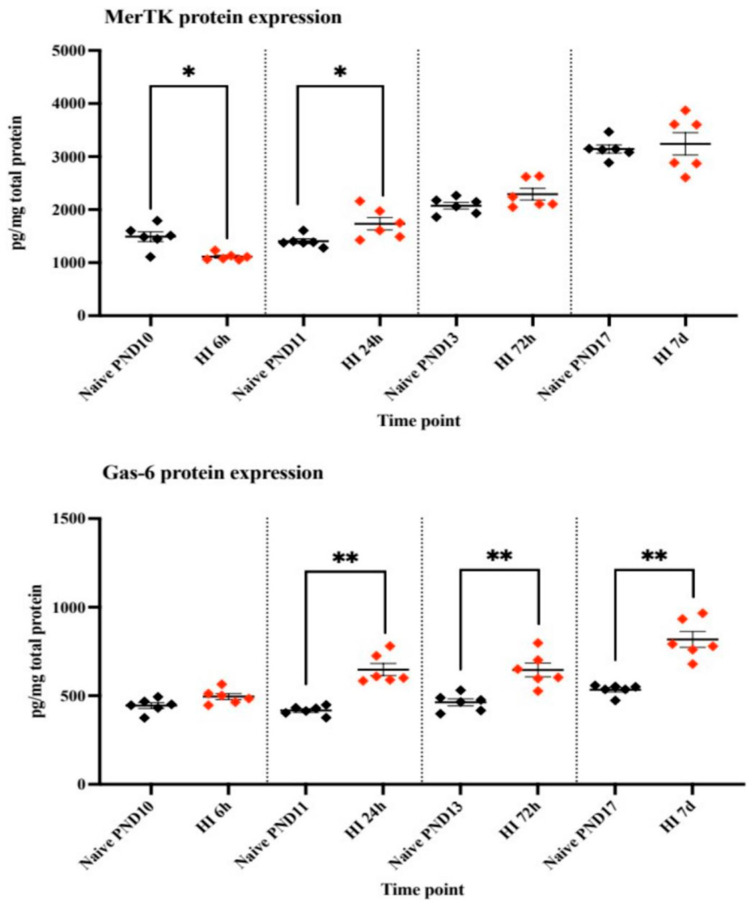
MerTK and Gas-6 protein expression in naïve animals and in animals after hypoxic-ischemia (HI). MerTK (myeloid-epithelial-reproductive tyrosine kinase), protein expression (ELISA) in naïve animals (black dots) and after HI (hypoxic-ischemia) (red dots) at different time points. Gas-6 (growth arrest–specific 6) protein expression (ELISA) in naïve animals (black dots) and after HI (red dots) at different time points. N = 6/group. Data is presented as individual animals and as mean ± SEM; (Mann–Whitney * *p* < 0.05, ** *p* < 0.01).

**Figure 5 cells-14-01862-f005:**
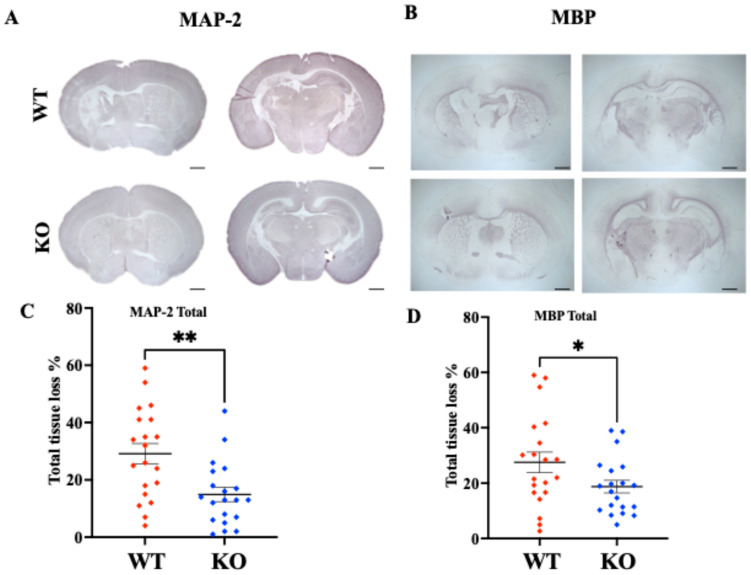
Tissue loss (%) in MerTK KO and WT mice 7 days after HI at different levels in the brain. (**A**) Representing whole brain micrographs in MerTK (myeloid-epithelial-reproductive tyrosine kinase), WT (wild type) and KO (knock-out) animals after staining with MAP-2 (microtubule associated protein—2). Levels showing striatum (**left**) and hippocampus/thalamus (**right**); (**B**) representing whole brain micrographs (×1.25) in WT and KO animals after staining with MBP (myelin basic protein). Levels represented are striatum (**left**) and hippocampus/thalamus (**right**); (**C**) graph represents the total tissue loss seen after staining with MAP-2 in WT vs. KO animals. Calculated as mean % area loss across 8 levels in the brain (** *p* = 0.002; *n* = 20/group). (**D**) Mean % area loss after staining for MBP in WT vs. KO animals (*n* = 20/group) representing the corpus callosum and external capsule, calculation based on area loss across 8 levels in the brain (* *p* = 0.04, (*n* = 20/group). Gender equally distributed in all analysis. Data presented as individual animals and as mean ± SEM and analyzed using D’Agostino and Pearson normality test followed by unpaired *t*-test. Tissue loss calculated as difference in area (mm^2^): (area ipsilateral–area contralateral–)/area contralateral × 100. * *p* < 0.05, ** *p* < 0.01.

**Figure 6 cells-14-01862-f006:**
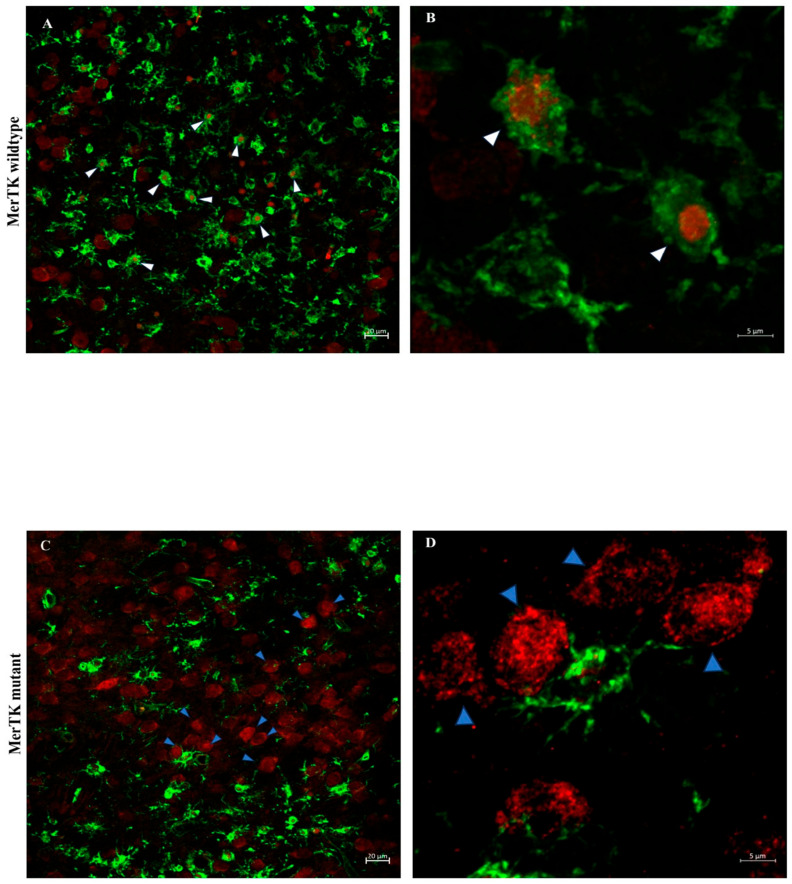
Confocal microscopy of neuronal (NeuN) and microglial immunoreactivity (isolectin) 72 h after HI in striatum. Quantification of 3D image orthogonal projections of confocal z-stacks of the injured area. (**A**,**B**) represents MerTK (myeloid-epithelial-reproductive tyrosine kinase), WT (wild type) mice, (**C**,**D**) MerTK KO (knock-out) mice. The number of microglia (green) containing neuronal nuclear (red) (NeuN) material is considerably reduced in MerTK KO animals after HI (KO: *n* = 13 and WT: *n* = 11). White arrows indicate neurons counted as engulfed, blue arrows indicating neurons not engulfed by microglia. (**A**,**B**) injured area in the striatum of the ipsilateral hemisphere in a WT animal. B same picture as A but with a higher magnification (×63) pointing at neurons being engulfed. (**C**,**D**) Injured area in striatum of the ipsilateral hemisphere in a MerTK KO animal. (**D**) Same picture as (**C**) but with a higher magnification (×63) showing neurons that are not localized inside microglia. Scale bar = 5 μm for picture (**B**,**D**) and 20 μm for picture (**A**,**C**).

**Figure 7 cells-14-01862-f007:**
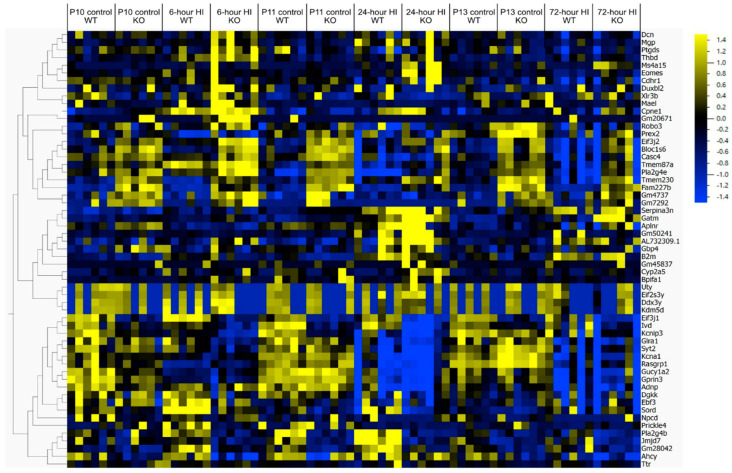
Heat-map of normalized counts from DeSeq analysis. Genes involved in phagocytosis, inflammation, and cell death in WT (wild type) vs. KO (knock-out) animals at 6 h, 24 h and 72 h after HI along with control animals. Samples ordered by animal group while genes by hierarchical clustering generating genes with similar expression patterns across groups close together. Yellow color indicates upregulation and blue color down regulation of genes.

**Figure 8 cells-14-01862-f008:**
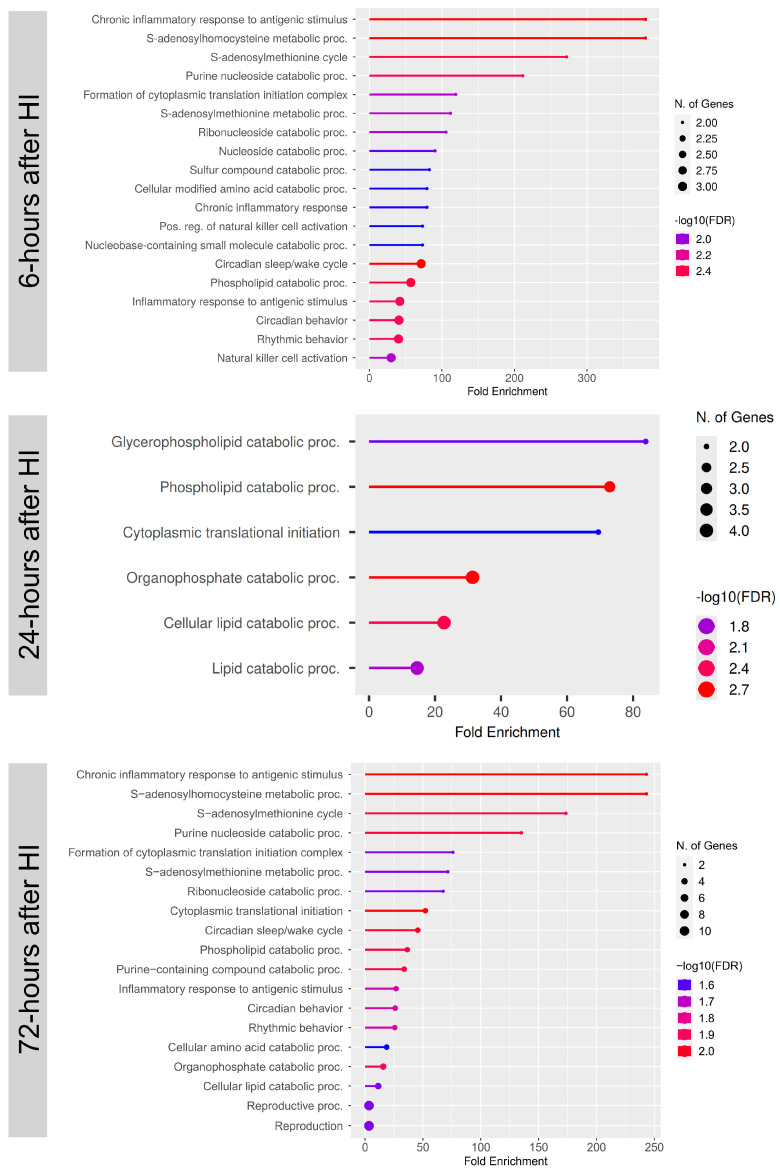
GO term analysis in MerTK KO vs. WT animals after HI. Figure displays a GO (gene ontology) term analysis of the most significant GO top biofunctions at 6 h (hour), 24 h, and 72 h after HI. *p* value < 0.05.

**Figure 9 cells-14-01862-f009:**
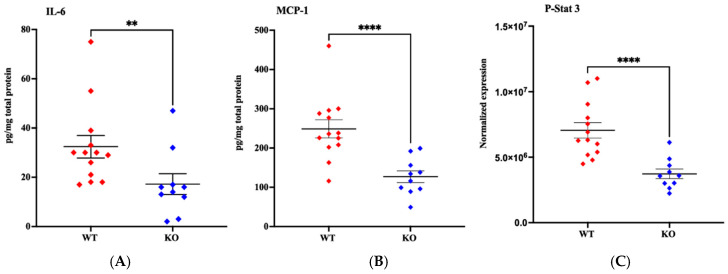
Protein levels of IL-6, MCP-1 and P-Stat3 in MerTK WT and KO animals after HI. Levels of IL-6 (interleukin-6) (**A**), MCP-1 (monocyte chemoattractant protein-1) (**B**) and P-Stat-3 (signal transducer and activator of transcription 3) (**C**) 6 h after HI in MerTK (myeloid-epithelial-reproductive tyrosine kinase WT (wild type) and MerTK KO (knock-out) mice. The concentration was expressed as pg/mg total protein for IL-6 and MCP-1 and as normalized expression for P-Stat 3. Expression in MerTK KO and WT were compared. WT-red dots, KO-blue dots. Data presented as individual animals ± SEM and analyzed using Mann–Whitney test. Levels of significance: ** *p* = 0.01; **** *p* < 0.0001.

## Data Availability

The original contributions presented in this study are included in the article. Further inquiries can be directed to the corresponding author.

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
