# Peer review of "MerTK and the Role of Phagoptosis in Neonatal Hypoxia-Ischemia"

_cells, 2025, doi:10.3390/cells14231862_

Round 1

Reviewer 1 Report

Comments and Suggestions for Authors

This study is well conducted, and the manuscript is well written overall.

I have some minor concerns on this manuscript.

Given-name is shown for one of authors, while initials of given-name are shown for other authors.

In the Abstract:

What is the meaning of “NeuN immunostaining in microglia” in the second paragraph in the Abstract.

Abbreviation, MerTK should be used in the last sentence of the second paragraph in the Abstract.

The final paragraph of the Abstract is not clear.

In the Methods:

In a subsection, Animals, authors describe “genotyping of each animal was after the experiments”, but in a subsection, Hypoxia-ischemia, they describe “MerTK KO mice (n=20) were exposed to HI”. How was it possible?

It is not clearly described which area of the brain was analyzed in each evaluation. The Vannucci model is known to have a large inter-animal variability with respect to tissue damage. Hence, it is crucial to clearly define and describe the analyzed areas in each evaluation, especially in immunohistochemistry.

In the Figures:

Resolution of Figure 8 is too low to recognize the results of the analysis.

In Figure 9, ** may indicate p <0.01, not p <0.001.

In the Discussion:

The authors consistently argue only microglia in the brain. After HI, however, invasion of macrophages into the brain parenchyma may occur. Discussion on this issue may clarify the role of MerTK.  

Reviewer 2 Report

Comments and Suggestions for Authors

The study provides mechanistic insight into early phagocytosis after neonatal HI. The results of the study are beneficial to broader research audience. The experiments are well executed, however overall flow of the writing can be improved. The study is missing the thought process why certain steps were chosen (see below). 

  • how was MerTK chosen as the primary focus of the study is not clear
  • the introduction can be improved by "jumping right in" as almost all publications start with HI is a big problem and we don't have treatments 
  • p2 Ln 56- cite 
  • p.3 Ln 105-106 merge and improve flow of the sentence 
  • p.4: comment on the timepoint choice 
  • why 7 cytokine Bioplex assay? 
  • Fig. 2 x-axis and y-axis can be slightly bigger 
  • Fig. 8 is blurry 

Reviewer 3 Report

Comments and Suggestions for Authors

Overall Evaluation

This manuscript provides important insights into hypoxic–ischemic brain injury during development. In particular, the demonstration of alterations in the MerTK–Gas6 axis, the regulation of phagoptosis-related molecular pathways, and the link between MerTK deficiency, reduced microglial engulfment, and neuroprotection is a major strength. The study benefits from the use of multiple complementary approaches, including RNA-seq, immunofluorescence, ELISA, and Western blotting, and the consistency across datasets increases the credibility of the findings.

The experimental data are generally sound; however, several issues need to be addressed regarding wording, methodological descriptions, figure quality, and consistency in terminology. In addition, the manuscript contains a number of spelling errors and inconsistencies in scientific terms. Please revise according to the comments below, and I strongly recommend a thorough professional English-language proofread of the full text.

Detailed Comments

Introduction

Line 83–89

Using the term “phagoptosis” in the context of early brain development may be scientifically inaccurate. During development, synaptic pruning is the dominant process. Although pruning shares some mechanistic features with phagoptosis, phagoptosis specifically refers to “phagocytic removal of viable but stressed neurons,” which is conceptually distinct.
→ It would be more accurate to refer to “excessive synaptic pruning” or “phagoptosis-like neuronal removal” rather than implying classical phagoptosis in the neonatal brain.

Figure 1

The primary ligand for P2Y6 is UDP, not ADP. P2Y6 responds very weakly to ADP.
→ Please correct the ligand in the schematic.

English corrections

  1. “the cellular mechanisms behind cell death … is yet not fully understood”
     are not yet fully understood
  2. “phagocytosis is mediated of the stressed neurons”
     “mediated by microglia” or “leading to phagocytosis”
  3. “the role of phagoptosis and MerTK and have not yet been investigated”
     Remove the extra and:
     the role of phagoptosis and MerTK has not yet been investigated
  4. “development of the immature brain (15) the molecular machinery …”
     Comma missing:
     (15), and the molecular machinery …

Methods

Animals (Line 92–103)

  • Please provide a brief description of the genotyping procedure.
  • Carotid ligation is a surgical intervention; therefore, anesthetic and analgesic protocols must be described.

Hypoxia-ischemia (Line 104–120)

  • Monitoring details during surgery (e.g., body temperature maintenance, mortality rate, perioperative care) should be added.

Sex distribution

  • Please justify the use of mixed sexes in the same analysis or provide rationale for not separating male and female pups.

RNA sequencing (Line 121–135)

  • Reporting RIN values (RIN ≥ 7) is essential for RNA-seq quality assessment.
  • Missing standard NGS parameters:
    • read depth
    • read length
    • single-end vs paired-end
      → These should be stated clearly.

Results

Line 261–263

  • Classifying P2Y12, GRO-1, C3a, and C5a as find-me signals is not fully accurate.
    → Consider describing them as “chemoattractants” or “microglial-activating molecules,” or provide additional references.
  • “Complement 3 receptor” should be written as CR3 (CD11b/CD18).
  • GD3 (ganglioside) is associated with dying-cell signaling, but is not a typical ‘eat-me signal’.
    → A more specific citation is needed.

Line 281

  • “controls animals” → control animals

Figure 5

  • WT and KO labels are partially cut off; figure revision is needed.

Line 324–326

  • “% microglia with NeuN” requires explicit description in Methods of how total microglial numbers were counted (definition of microglia, counting strategy, thresholds).

Figure 8 and Figure 9

  • The resolution is too low; text is unreadable.
    → Please replace with high-resolution figures.
  • Figure numbering does not match the text in certain places; please unify.

Discussion

Line 383–385

  • “also are regulated” → are also regulated

Line 411–414

  • The correct reference should likely be Figure 3, not Figure 1.

Line 441–443

  • SOCS3 was not measured at the protein or ELISA level in Methods/Results.
    → If referring to RNA-seq data, state this explicitly; otherwise, revise or remove to avoid confusion.

Line 421–424

  • There are formatting issues with excess spacing.
  • The argument based on Figure 7 is highly speculative; specific genes and their directional changes should be cited or the wording softened.

Abbreviations

Please correct the following:

  • EI3JA → EIF3A
     (Eukaryotic translation initiation factor 3 subunit A)
  • micro tubule → microtubule
  • signal tranducer → signal transducer
  • vitro nectin receptor → vitronectin receptor
  • intra peritoneally → intraperitoneally
  • PCA → Principal Component Analysis
